# Factors Affecting Employees Work Engagement in Small and Medium-Sized Enterprises

**DOI:** 10.3390/ijerph191710702

**Published:** 2022-08-27

**Authors:** Weng Li, Yasmin Jahan, Madoka Kawai, Yasuko Fukushima, Kana Kazawa, Michiko Moriyama

**Affiliations:** 1Division of Nursing Science, Graduate School of Biomedical and Health Sciences, Hiroshima University, Hiroshima 734-8553, Japan; 2Department of Medicine for Integrated Approach to Social Inclusion, Graduate School of Biomedical and Health Sciences, Hiroshima University, Hiroshima 734-8553, Japan

**Keywords:** work engagement, health literacy, small and medium-sized enterprises, job satisfaction, health and productivity management

## Abstract

Background: Improving the labor productivity of small and medium-sized enterprises (SMEs) is essential because of the aging population and predicted reduction in the labor force. Therefore, it is necessary to ensure that employees are in good health to work for long times. In this regard, the purpose of the study is to investigate the relationship between work engagement (WE) and related variables among SME employees. Methods: A cross-sectional study was conducted using the baseline data of the prospective cohort study, which included 377 employees from three SMEs headquartered in Hiroshima Prefecture, Japan. Results: A multiple regression analysis was performed to see the associations between Utrecht Work Engagement Scale (UWES) scores and other variables. Significant associations were found with job satisfaction, age, health literacy (HL), and quality of sleep (all, *p* < 0.05). Additionally, there was a statistically significant difference observed in WE, quality of sleep, concern for own body, job satisfaction, and family life satisfaction (all, *p* < 0.001) with respect to high and low HL levels. Conclusions: The results of this study reveal that while implementing health and productivity management in SMEs to increase WE, it is best to concentrate on raising HL, job satisfaction, and sleep quality. To increase generalizability, further research could be conducted with more SMEs companies.

## 1. Introduction

With the aging population, the impact of declining birthrates, and the reduction in the labor force in the future, it is important to improve the labor productivity of small- and medium-sized enterprises (SMEs), which account for 99.7% of all companies in Japan. Especially, SMEs often face challenges in retaining qualified personnel and recruiting new staff. Therefore, it is necessary to ensure that current employees can work for a long time in good health. For this purpose, the Japanese government has introduced the health and productivity management (HPM) concept to SMEs following large industries to increase labor productivity and enhance the performance of their strategic policy [1].

In this context, work engagement (WE) is gaining attention as an organizational management indicator in HPM. The WE is a positive, satisfying work-related state of mind characterized by vigor, dedication, and immersion [2]. In recent years, a meta-analysis has reported [3] that low WE may contribute to decreased well-being and work performance. Some research found significant relationships between WE, job satisfaction [4], job performance, and retention [5,6,7].

The WE is mostly related to health problems. Previous studies explored various behavioral factors, such as sleep hygiene, that may influence job engagement [8]. Quality of sleep appears to be a key predictor of employees’ behavior and experience at work, including job engagement [9]. Recovery from work (i.e., psychological detachment, relaxation, mastery, and control) is positively related to both WE and quality of sleep [10]. Further, there is a link between life satisfaction, work–life balance, and WE [11]. Job–life balance has a positive relationship with both life and works satisfaction [12], and life satisfaction increases WE. Even while motivated workers are more productive in the modern workplace, many people still fail to see the importance of work–family and work–life balance [13].

Health literacy (HL) is also discussed as a key factor for good health [14]. HL is defined by the World Health Organization as “the cognitive and social skills which determine the motivation and ability of individuals to gain access to, understand, and use information in ways which promote and maintain good health” [15]. Studies found that high HL can improve workers’ health [16,17], and low HL can limit the understanding and training effectiveness of occupational health and safety [18]. HL has the potential to reduce health problems and maintain long-term health, lead to a healthier work–life via health practices, and further may prevent workers from leaving their jobs because of health reasons. 

Nowadays, none of the industries can survive without considering employees’ WE. Thus, it is critical to understand the concept of WE and its implications for SMEs. Till now, numerous studies confirm the importance of job satisfaction, work–family balance, life satisfaction, quality of sleep, and WE, but there is a lack of research that explores the relationship between all these concepts together WE. As a result, we decided to conduct a project to improve the WE of SMEs in Japan, and a framework would be required to carry it out [19,20]. Therefore, this study proposed a research model (Figure 1) that aims to analyze the relationship between WE and related variables among SME employees. Based on that, this research hypothesized that: 

**Hypothesis** **1.***Sociodemographic variables such as sex, age, marital status, position/job type, and physical activity level may have a significant influence on WE among SME employees*.

**Hypothesis** **2.***Concern for own body, job satisfaction, family life satisfaction, quality of sleep, and HL may have a significant influence on WE among SME employees*.

Consequently, the result will give us a direction on how to improve the WE of the trial companies based on the different characteristics.

## 2. Materials and Methods

### 2.1. Study Design, Setting, and Sample

The following was the baseline data of a prospective cohort study in which each participating company implemented HPM, and the researchers observed the chronological changes of WE and other outcome variables of the companies. The researcher obtained written informed consent from the president of each company to participate in this study. The personnel manager informed the employees by in-house message or e-mail and asked about their participation in this study. Employees who did not want their data to be used for this study were told to report. Therefore, data analysis was conducted on an opt-out basis.

Self-administered questionnaires were conducted between August and November 2019. Participants were from three SMEs headquartered in Hiroshima Prefecture, the western part of Japan—those who were willing to participate in this study; Company A (wholesale), Company B (supermarket), and Company C (transportation (taxi)). Different industries were purposively selected because the industrial reports [21] indicated that there was a large difference in the health status of employees.

### 2.2. Questionnaire and Measures

We collected sociodemographic data such as sex, age categories, marital status, years of employment, and position/job type. Physical activity level was classified by the Ministry of Health, Labour, and Welfare (2006). Classification levels are 1 = low (mostly sitting), 2 = moderate (regular normal activities), and 3 = high (heavier activities/active movement) [22]. The WE was measured using the Japanese version of the Utrecht Work Engagement Scale (UWES). The UWES is a nine-item self–administered scale consisting of three subscales with three items each [23]. All items are scored and evaluated on a seven-point rating scale ranging from 0 (never) to 6 (always), and divide the total score of the answers to 9 items by 9 to get the score. The higher the score, the higher the engagement. HL was measured by the Japanese version of the European Health Literacy Survey Questionnaire (HLS-EU-Q47) [24]. It is a 12-dimensional scale, and each item was rated on a four-point Likert scale (very easy, slightly easy, slightly difficult, and very difficult), ranging from 0 to 50 points. The higher the score, the higher HL. HL was categorized into the following four levels based on total score: “inadequate”, 0–25 points; “problematic”, >25–33 points; “sufficient”, >33–42 points; “excellent”, >42 points or more. “Inadequate” and “problematic” HL scores with 33 points or less were defined as “limited” in this study. Quality of sleep was assessed by Pittsburgh Sleep Quality Index (PSQI). The PSQI consists of 18 questions which generate seven subcomponent scores, each ranging from 0 to 21 (0 score equals better and 21 is worst), and 5.5 points are set as the cutoff value [25]. The degree of concern for his/her body was evaluated using an 11-point Likert scale in which “very concerned” was 10 and “not concerned at all” was 0. A higher score represented a higher degree of concern for own body. Job satisfaction and family life satisfaction are measured by the Brief Job Stress Questionnaire (BJSQ) [26]. The BJSQ consists of 57 items used to assess job stressors, and the item score of the four-point Likert scale questions ranging from “satisfied with work” and “satisfied with family life” was extracted, with 1 (satisfied), 2 (somewhat satisfied), 3 (slightly dissatisfied), and 4 (dissatisfied). The lower the score, the higher the satisfaction level.

### 2.3. Data Analysis

For data analysis, statistical software SPSS (ver. 25, IBM, Armonk, NY, USA) was used, and the significance level was set to less than 5%. For multiple analysis, the significance level was determined by several factors. Descriptive statistics were performed on the basic characteristics and the scores of each scale. For each scale and each subscale, Cronbach’s alpha coefficient was calculated to confirm reliability. The mean and standard deviations (SD) of each variable (UWES, PSQI, HLS-EU-Q47, concern for own body, job satisfaction, and family life satisfaction) were calculated. 

First, we analyzed the characteristics of the entire group of participants. Second, to compare the basic characteristics of the three companies, a chi-squared test, and Kruskal–Wallis test was performed. The age was further categorized into three groups (<30, 30–59, and 60 years and over), years of employment were categorized into two groups (<3 years, 3 years or over), and position and job type were in two groups (managers and others).

To see the effect on WE, the nominal scale (gender, marital status) was changed to a dummy variable, and Pearson’s correlation analysis was performed. In addition, a stepwise multiple regression analysis was performed with the UWES score as a dependent variable and other items as independent variables. To assess the impacts of HL levels on other variables, a *t*-test was performed for the high HL and the low HL groups, and the Mann–Whitney U test was performed for concern for own body, job satisfaction, and family life satisfaction for the Likert scale after confirming the data normality.

To compare the differences between the three companies, each scale was used as a dependent variable, and each company was used as the explanatory variable. The Kruskal–Wallis test was performed to test for differences between the concern for own body, job satisfaction, and family life satisfaction (ordinal scale) where appropriate (if data were not normally distributed). One-way ANOVA and multiple comparisons using Tukey’s methods were performed after verification of data normality. 

### 2.4. Ethics Statement

This study was approved by the epidemiological research ethics committee of Hiroshima University (E-1502-1-E1502-3, UMIN000036231). After collecting the data, the third person from any company deidentified the personal information, assigned unique numbers, and sent the data to the researchers.

## 3. Results

### 3.1. Sociodemographic Factors of Employees and Comparison of Variables among Companies

A total of 760 employees were enrolled, and 439 of them returned (the return rate was 57.8%). Three hundred seventy-seven valid answers were analyzed for this study (the valid response rate was 85.9%). The sociodemographic characteristics of the employees are shown in Table 1. More males (60.5%), and the highest percentage of age group was in the 40′s (27.6%) following 50′s (26.5%). The majority (70.3%) were married, and the most common years of employment were 10 to 15 years (21.8%). Job types were the highest clerical work (Company A), sales (Company B), and driving (Company C), respectively. Physical activity is strongly related to work types, and it showed that 17% were low (Level 1), 45.1% were normal (Level 2), and 33.4% were high (Level 3).

When comparing characteristics between companies, Company A (wholesales) and C (taxi) had a large proportion of males (over 80%), and Company B (supermarket) had a large proportion of females (84.4%) (*p* < 0.001). Age is significantly higher in Company C. There were no significant differences among the three companies in terms of marital status and years of employment. Physical activity is significantly lower in Company C (*p* < 0.001).

When comparing WE, HL, quality of sleep, concern for own body, job satisfaction, and family life satisfaction among three companies by multiple comparisons (Table 2), the WE score was significantly higher for Company C than for Companies A and B (*p* < 0.001). Regarding the quality of sleep score, Company C was significantly lower than Companies A and B (both, *p* < 0.001), which indicated that employees of the taxi company had shown a better quality of sleep. There were no significant differences between the three companies in terms of HL, the concern for own body, work satisfaction, and family life satisfaction.

Among seven scales, two scales (quality of sleep and HL) have cutoff points. On PSQI, 39.0% have sleep disorders with a score of 6 or higher, and on HLS-EL-Q47, 72.4% showed low HL with a score of 33 or lower. Company C showed the lowest sleep disorder rate among the three companies (*p* < 0.001). Regarding HL, no significant differences were observed among the three companies (Table 3).

### 3.2. Multiple Regression Analysis with WE as a Dependent Variable

Table 4 shows the result of multiple regression analysis (stepwise method) for WE measured by UWES as a dependent variable. Those showing significant associations were job satisfaction (β = −0.38, *p* < 0.001), age (β = 0.28, *p* < 0.001), HL (β = 0.14, *p* < 0.001), and quality of sleep (β = −0.11, *p* = 0.01). The results showed that higher job satisfaction, higher age, higher HL, and a better quality of sleep contributed to WE.

### 3.3. Impacts of HL on Other Variables among three Companies

Table 5 shows the impacts of HL on other variables among the three companies. There was a statistically significant difference in WE, quality of sleep, concern for own body, job satisfaction, and family life satisfaction (all, *p* < 0.001). The high HL group showed statistically significant improvement in all variables.

## 4. Discussion

To our best knowledge, this is the first study where we have shown the associations between WE and other variables. HL is also important for improving WE, quality of sleep, and job and family life satisfaction, even though β was not high. However, based on occupation, it was revealed that the age group and gender ratio varied. The national statistics for Japan (statistics bureau of Japan, 2021) and other developed countries show similar trends [27,28].

The mean value of WE in the total sample of this study was 3.32 (SD 1.13), which was the same as Sakurai et al. [29] surveyed Kindergarten teachers. When compared with previous Japanese research [30,31,32], in which the mean value of the nurse survey was 2.68 points, dairy employees were 2.9 points, and SME employees were 2.92 points, participants of this study showed higher WE. This could be affected by the company’s higher motivation to participate in this study. The WE of the taxi company (3.32), which had the largest proportion of middle-aged and older employees, showed the highest mean value in this study. The previous studies [33,34,35] revealed that the WE increased with age. With this study result, once the proper quantity of work is adjusted and the quality of sleep is maintained, employees who have retired and are rehired feel more engaged at WE because they believe working is worthwhile. This might strengthen the SMEs which are facing aging employees.

Surprisingly, the taxi company showed better conditions regarding sleep status. Sleep disorders and attention deterioration are physiological effects of aging [36,37], and according to the reports, traffic accidents by aged drivers are on the rise [38,39]. The transportation companies’ attention to their employees and the improvement of the work environment [40] may help them sleep better. In fact, this company makes an HPM declaration and cares about the health of its employees. On the other hand, taxi drivers significantly engaged with the lowest physical activity. Therefore, it is important to pay attention to the onset of disease, and transportation companies should take steps to boost their employees’ physical activity.

This study result showed a high proportion of low HL among the companies. The total HL of the three companies was 27.74 ± 9.58, which was about 2 points higher than the results of other surveys conducted in Japan [25]. It is worth noting that Japan’s HL scores are lower than those of the other six Asian countries (about 30 points) [41] and European countries (33.8 points) [42]. The reasons for this gap may be addressed in Japan’s insufficient primary care system, which is a health education platform for all citizens, and a lack of an easy-to-understand and reliable, comprehensive internet site for general citizens such as MedlinePlus [25]. There are well-developed governmental information sites in Singapore [43] and Sweden [44]. This high proportion of low HL is concerning, given the important influence of HL on all variables assessed in this study. The HPM project may use this data as an intervention.

This study also found that the higher the HL, the higher the WE, the better the sleep condition, and the higher the degree of physical concern, degree of job satisfaction, and degree of family life satisfaction. These results were supported by the previous studies that women with high HL showed better work performance, and their positive health behavior led to preventative health measures and investment in health, which improved and maintained workers’ health while also enhancing their work performance [45,46]. In addition, when sleep condition deteriorates, WE decrease [47,48]. These results proved that HL played a key role in improving health behavior. As a result, offering health education to increase HL is critical for improving work performance and productivity.

This study result showed that there was no connection between WE and family life satisfaction. In the previous study, WE had a positive correlation with life satisfaction (satisfaction with work and family) [49,50]. The results of this study showed that increased job satisfaction is crucial for raising engagement in SMEs.

### Limitations

This research has a few limitations:(1)We used the baseline data of the cohort study, and we selected one company from each industry. Therefore, it is difficult to conclude the characteristics of each industry. In the future, multiple companies could be involved for comparison;(2)Since this is a cross-sectional design, no causal relationship can be established. On the other hand, intervention research will be required to identify the key variables by examining the correlations between variables;(3)We cannot generalize this study’s findings because of the small sample size and purposefully selected three SMEs. Therefore, there is a chance of selection bias.

## 5. Conclusions

An analysis was conducted to clarify the relationship between employees’ WE, concern for their own bodies, quality of sleep, family life satisfaction, and job satisfaction in SMEs. The results showed that WE was affected in job satisfaction, age, HL, and quality of sleep. We also found the impacts of HL on other variables among the three companies. Therefore, it is suggested to focus on increasing HL, job satisfaction, and improving the quality of sleep when implementing HPM in SMEs to improve WE. Additionally, it can be suggested that further research can be conducted with more SMEs company in order to boost generalizability.

## Figures and Tables

**Figure 1 ijerph-19-10702-f001:**
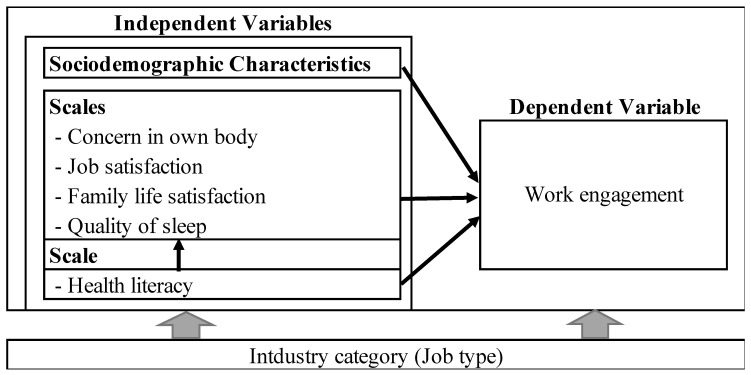
A Conceptual framework of this study.

**Table 1 ijerph-19-10702-t001:** Sociodemographic characteristics of the study participants of three companies (*n* = 377).

		Total	Company A (*n* = 98)	Company B (*n* = 135)	Company C (*n* = 144)	
Variables		*n* (%)	*n* (%)	*n* (%)	*n* (%)	*p*
Sex						
	Male	228 (60.5)	80 (81.6)	21 (15.6)	127 (88.2)	<0.001 ☨
	Female	149 (39.5)	18 (18.4)	114 (84.4)	17 (11.8)	
Age group (year)						
	<20	3 (0.8)	0	3 (2.2)	0	<0.001 ☆
	20–29	20 (5.3)	9 (9.2)	10 (7.4)	1 (0.7)	
	30–39	54 (14.3)	29 (29.6)	20 (14.8)	5 (3.5)	
	40–49	104 (27.6)	34 (34.7)	41 (30.4)	29 (20.1)	
	50–59	100 (26.5)	17 (17.3)	34 (25.2)	49 (34.0)	
	60–69	76 (20.2)	9 (9.2)	27 (20)	40 (27.8)	
	70–79	20 (5.3)	0	0	20 (13.9)	
Marital status						
	Single	112 (29.7)	36 (36.7)	39 (28.9)	37 (25.7)	0.176 ☨
	Married	265 (70.3)	62 (63.3)	96 (71.1)	107 (74.3)	
Years of employment						
	<1	39 (10.3)	8 (8.2)	14 (10.4)	17 (11.8)	0.659 ☆
	<3	51 (13.5)	12 (12.2)	15 (11.1)	24 (16.7)	
	<6	65 (17.2)	12 (12.2)	25 (18.5)	28 (19.4)	
	<10	57 (15.1)	12 (12.2)	17 (12.6)	28 (19.4)	
	<15	82 (21.8)	21 (21.4)	45 (33.3)	16 (11.1)	
	<20	36 (9.5)	14 (14.3)	10 (7.4)	12 (8.3)	
	≥20	47 (12.5)	19 (19.4)	9 (6.7)	19 (13.2)	
Position/Job type						
	Manager	45 (11.9)	25 (25.5)	4 (3.0)	16 (11.1)	<0.001 ☨
	Engineer	6 (1.6)	1 (1.0)	5 (3.7)	0	
	Clerical work	41 (10.9)	26 (26.5)	1 (0.7)	14 (9.7)	
	Sales	74 (19.6)	14 (14.3)	59 (43.7)	1 (0.7)	
	Service	62 (16.4)	4 (4.1)	33 (24.4)	25 (17.4)	
	Production	13 (3.4)	0	0	0	
	Mailing/Mechanical driving	71 (18.8)	9 (9.2)	13 (9.6)	62 (43.1)	
	Production process	6 (1.6)	0	6 (4.4)	0	
	Transportation/Cleaning/Packaging	16 (4.2)	8 (8.2)	5 (3.7)	3 (2.1)	
	Others	33 (8.8)	9 (9.2)	9 (6.7)	15 (10.4)	
	Missing value	10				
Physical activity level						
	level 1	64 (17.0)	12 (12.2)	6 (4.4)	46 (31.9)	<0.001 ☆
	level 2	170 (45.1)	37 (37.8)	46 (34.1)	87 (60.4)	
	level 3	126 (33.4)	44 (44.9)	79 (58.5)	3 (2.1)	
	Missing value	17	5	4	8	

☨: Chi-square test, ☆: Kruskal–Wallis. Company A, wholesale; Company B, supermarket; Company C, taxi.

**Table 2 ijerph-19-10702-t002:** The score of participant scales and its comparison between three companies.

Company	Work Engagement	Health Literacy	Quality of Sleep	Concern for Own Body ^b^	Job Satisfaction	Family Life Satisfaction
	UWES^a^	HLS-EU-Q47 ^b^	PSQI ^b^	BJSQ ^b^
	Mean ± SD	Mean ± SD	Mean ± SD	Mean ± SD	Mean ± SD	Mean ± SD
A: *n* (98)	3.12 ± 1.07	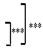	26.50 ± 10.10	n.s.	5.61 ± 2.60	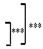	7.35 ± 2.25	n.s.	2.16 ± 0.74	n.s.	1.78 ± 0.75	n.s.
B: *n* (135)	3.12 ± 1.05	28.18 ± 9.05	5.58 ± 2.91	7.36 ± 2.11	2.27 ± 0.84	1.90 ± 0.92
C: *n* (144)	3.64 ± 1.18	28.18 ± 9.69	4.03 ± 2.56	6.97 ± 2.26	2.12 ± 0.83	1.88 ± 0.84
Total: 377	3.32 ± 1.13		27.74 ± 9.58		4.99 ± 2.80		7.20 ± 2.21		2.18 ± 0.81		1.86 ± 0.85	
Cronbach’s α	0.74		0.85	0.76	-	-	-

(a): Multiple comparisons (Tukey) (b): Kruskal–Wallis, *** *p* < 0.001, n.s.—not significant. A (Company = A), wholesale; B (Company B), supermarket; C (Company C), taxi. UWES—Utrecht Work Engagement Scale. The higher the score, the higher the engagement. PSQI—Pittsburgh Sleep Quality Index. The lower the score, the higher the Quality of sleep. The cutoff point is 5.5. HLS-EU-Q47—European Health Literacy Survey Questionnaire. The higher the total score, the higher the health literacy. Concern for own body: The higher the score, the higher the satisfaction level. BJSQ—Brief Job Stress Questionnaire. The lower the score, the higher the satisfaction level.

**Table 3 ijerph-19-10702-t003:** Comparison of health literacy levels and sleep disorders among three companies.

	Total	Company A (*n* = 98)	Company B (*n* = 135)	Company C (*n* = 144)	
Variables	*n* (%)	*n* (%)	*n* (%)	*n* (%)	*p*
Quality of sleep (PSQI)					
≤5	230 (61.0)	49 (50.0)	74 (54.8)	107 (74.3)	<0.001 ☨
>5	147 (39.0)	49 (50.0)	61 (45.2)	37 (25.7)	
Health literacy (HLS-EU-Q47)					
≤33	273 (72.4)	73 (74.5)	96 (71.1)	104 (72.2)	0.848 ☨
>33	104 (27.6)	25 (25.5)	39 (28.9)	40 (27.8)	

☨ Chi-square test. Company A, wholesale; Company B, supermarket; Company C, taxi. PSQI: Pittsburgh Sleep Quality Index. The lower the score, the higher the Quality of sleep. The cutoff point is 5.5. HLS-EU-Q47: European Health Literacy Survey Questionnaire. The higher the total score, the higher the health literacy.

**Table 4 ijerph-19-10702-t004:** Stepwise multiple regression analysis for Work Engagement.

Variables	RC	SRC	95% CI		*p*
Job satisfaction	−0.53	−0.38	−0.65	−0.4	<0.001
Age	0.25	0.28	0.18	0.32	<0.001
HLS-EU-Q47	0.02	0.14	0.01	0.03	<0.001
PSQI	−0.05	−0.11	−0.08	−0.01	0.010

HLS-EU-Q47—European Health Literacy Survey Questionnaire. PSQI—Pittsburgh Sleep Quality Index.

**Table 5 ijerph-19-10702-t005:** Impacts of health literacy on other variables among three companies.

HL Scores	UWES ^a^	PSQI ^b^	Concern for Own Body ^b^	Job Satisfaction ^b^	Family Life Satisfaction ^b^
	Mean ± SD	Mean ± SD	Mean ± SD	Mean ± SD	Mean ± SD
HL score ≤ 33	3.16 ± 1.07	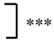	6.32 ± 2.87	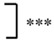	6.96 ± 2.11	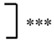	2.27 ± 0.79	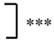	1.93 ± 0.83	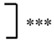
HL score > 33	3.73 ± 1.19	4.14 ± 2.43	7.84 ± 2.34	1.96 ± 0.82	1.66 ± 0.86

(a): *t*-test, (b): Mann–Whitney U test. *p*-value: *** *p* < 0.001. Company A, wholesale; Company B, supermarket; Company C, taxi. Work engagement: UWES; Quality of sleep: PSQI; HL: HLS-EU-Q47.

## Data Availability

Data used in this work can be obtained from the corresponding author upon request.

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
