# Peer review of "Factors Affecting Employees Work Engagement in Small and Medium-Sized Enterprises"

_ijerph, 2022, doi:10.3390/ijerph191710702_

Round 1

Reviewer 1 Report

The theme of this paper is interesting and innovative, and certainly useful.

This paper could be very interesting for researchers looking for this Journal.

However, this paper has some weaknesses.

ABSTRACT

The abstract is what attracts (or does not) the attention and interest to the article. So, it should be carefully written.

It is suggested that the authors retired all statistical data (e.g. β = -0.38, p <0.001).

1.INTRODUCTION

An introduction should be informative and well-worded. Therefore, it is missing:

-  insert the references to substantiate the text

-  present the theoretical problem/formulation and the objective of the study.

-  give clues to the discussion of the results.

-  present the structure of the article.

2. The paper does not display theoretical formulation. This should be concise, but not reductive.

It should present a review of the recent literature on the topics addressed and present the basic concepts.

In relation to the theme of the job satisfaction, there are more current references to be inserted. See, for example, the following papers:

Baquero, Delgado, Escortell, Sapena. (2019) Authentic Leadership and Job Satisfaction: A Fuzzy-Set Qualitative Comparative Analysis (fsQCA). Sustainability 11:8, 2412. https://doi.org/10.3390/su11082412

José Álvarez-García, María de la Cruz Del Río-Rama, Margarida Saraiva & António Ramos-Pires (2016) Dependency relationships between critical factors of quality and employee satisfaction, Total Quality Management & Business Excellence, 27:5-6, 595-612, DOI: 10.1080/14783363.2015.1021243

Margarita de Miguel-Guzmán, Alexander Sánchez-Rodríguez, Reyner Pérez-Campdesuñer, Gelmar García-Vidal, Rodobaldo Martínez-Vivar. (2018) Structure of the Variables that Affect Job Satisfaction Study in Cuban Organizations. International Journal of Management Science and Business Administration 4:4, 18-27. DOI: 10.18775/ijmsba.1849-5664-5419.2014.44.1002

María de la Cruz Del Río-Rama, José Álvarez-García, Margarida Saraiva & António Ramos-Pires (2017) Influence of quality on employee results: the case of rural accommodations in Spain, Total Quality Management & Business Excellence, 28:13-14, 1489-1508, DOI: 10.1080/14783363.2016.1150171

All this must be greatly improved.

2. MATERIAL AND METHODS

It shall provide the necessary and sufficient information to assess how the study was conducted in order to allow its reproduction by other.

All this relevant information is missing.

There are too many subpoints that must be eliminated

3. RESULTS

This point should be articulated with the previous one, as the results presented should be supported by the descriptive methods.

The results should show the evidence of the study and should be presented according to a logical and informative and perceptible sequence.

More than describing the tables and data, it is necessary to reflect and discuss the results obtained.

This point should be improved.

LIMITATIONS

I don't understand because this point is not in the conclusions!

5. CONCLUSION

The results should be presented in a comprehensive manner, highlighting the most relevant ones and provide a summary of the text.

The future investigation need referred here.

The originality and relevance of the results presented should be strengthened.

The limitations of the study itself should be presented and discussed here and not at a previous point

Author Response

Response to the Reviewer 1

Thank you very much for your valuable comments and suggestions. The authors have tried to follow your instructions in the comments that are listed below:

ABSTRACT

The abstract is what attracts (or does not) the attention and interest to the article. So, it should be carefully written.

 Response

The authors revised it accordingly.

It is suggested that the authors retired all statistical data (e.g. β = -0.38, p <0.001).

Response

Thank you very much for your valuable suggestions. After reviewing a few articles from MDPI journals, all of the authors agreed to maintain the statistical information as it is except for the β values. The authors eliminated the β values from the abstract.

  1. INTRODUCTION
    An introduction should be informative and well-worded. Therefore, it is missing: - insert the references to substantiate the text
    - present the theoretical problem/formulation and the objective of the study.
    - give clues to the discussion of the results.
    - present the structure of the article.

Response

The authors have inserted the suggested references and presented the theoretical framework accordingly.

We cited reference no. 20,21 in the text. Please see the 5th paragraph of the introduction section.

The article's structure and clues to the discussion are probably understandable because the author's main goal is to evaluate the relationship between WE and related characteristics among SME employees. The authors asked the reviewer to take this into consideration.

  1. The paper does not display theoretical formulation. This should be concise, but not reductive.

It should present a review of the recent literature on the topics addressed and present the basic concepts.

In relation to the theme of the job satisfaction, there are more current references to be inserted. See, for example, the following papers:

Baquero, Delgado, Escortell, Sapena. (2019) Authentic Leadership and Job Satisfaction: A Fuzzy-Set Qualitative Comparative Analysis (fsQCA). Sustainability 11:8, 2412. https://doi.org/10.3390/su11082412

José Álvarez-García, María de la Cruz Del Río-Rama, Margarida Saraiva & António Ramos- Pires (2016) Dependency relationships between critical factors of quality and employee satisfaction, Total Quality Management & Business Excellence, 27:5-6, 595-612, DOI: 10.1080/14783363.2015.1021243

Margarita de Miguel-Guzmán, Alexander Sánchez-Rodríguez, Reyner Pérez-Campdesuñer, Gelmar García-Vidal, Rodobaldo Martínez-Vivar. (2018) Structure of the Variables that Affect Job Satisfaction Study in Cuban Organizations. International Journal of Management Science and Business Administration 4:4, 18-27. DOI: 10.18775/ijmsba.1849-5664-5419.2014.44.1002

María de la Cruz Del Río-Rama, José Álvarez-García, Margarida Saraiva & António Ramos- Pires (2017) Influence of quality on employee results: the case of rural accommodations in Spain, Total Quality Management & Business Excellence, 28:13-14, 1489-1508, DOI: 10.1080/14783363.2016.1150171

All this must be greatly improved.

Response

The authors appreciate your efforts and recommendations. We read the articles carefully and displayed theoretical formulation accordingly with appropriate citations.

  1. MATERIAL AND METHODS

It shall provide the necessary and sufficient information to assess how the study was conducted in order to allow its reproduction by others. All this relevant information is missing.

Response

The authors added the necessary information accordingly.

There are too many subpoints that must be eliminated

Response

The authors have eliminated subpoints as suggested.

  1. RESULTS

This point should be articulated with the previous one, as the results presented should be supported by the descriptive methods. The results should show the evidence of the study and should be presented according to a logical and informative and perceptible sequence.

Response

The authors have tried to articulate the previous one and also added the descriptive methods accordingly. The authors updated the tables to make the results clearer.

More than describing the tables and data, it is necessary to reflect on and discuss the results obtained. This point should be improved.

Response

The authors have discussed the obtained results as the reviewer suggested.

LIMITATIONS
I don't understand because this point is not in the conclusions!

Response

Thank you very much for your valuable comments. The authors have added it in the conclusion section as suggested.

  1. CONCLUSION

The results should be presented in a comprehensive manner, highlighting the most relevant ones and provide a summary of the text.

The future investigation need referred here.

Response

The authors have done accordingly as suggested.

The originality and relevance of the results presented should be strengthened.

The limitations of the study itself should be presented and discussed here and not at a previous

Response

The authors have done accordingly.

Reviewer 2 Report

The manuscript Ijerph-1826776 entitled "Factors affecting employees work engagement of small and medium enterprises" analyses the relationship between work engagement and health literacy with work and family variables that influence them and thus health management and productivity. The attempt to integrate and relate all the variables that influence productivity in the "mother" concepts of WE and HL in the normative framework of HPM is a value to be highlighted, although this attempt is presented as ambiguous throughout the manuscript. 

 The study is adequately designed and developed, and the statistical analysis performed is adequate to achieve the objectives. The material and methods, which precisely lists the variables and the measuring instruments, but is imprecise in the description of the method and in the determination of the type of study. The results are presented in a clear and orderly manner and the discussion is adequate. 

 Nevertheless, a "mayor revision" of the present version is recommended before publication.

Specific comments:

Title: not adequate. It is recommended that the title be rewritten to reflect the two key aspects, work engagement (WE) and  Health literacy (HL) . 

0- Abstract: to be rewritten to adapt the suggestions in the manuscript 

1- Introduction:

It is brief and adequate, emphasising and defining the essential concepts of the study. 

However, although the definition made in the introduction of the LH is clear and relevant, the importance and its treatment as a dependent variable in the study is not sufficiently defined.

In L66 mention is made of "the relationship between all these concepts together with HL and WE" and in the results it is presented in section 2.3.3. Relationship between levels of HL and other variables (L158). 

Although the value that is intended to be given to HL (at the same level as WE) is intuited in the definition of the objective, HL is diluted, it disappears, there is no mention of it, only of WE, 

It is necessary to rewrite the definition of the OBJECTIVE to include this variable (HL). 

2. Methods:

2.1 Study design, setting and sample. 

L73 where it says "prospective observational cohort study" it is recommended to review and, if necessary, modify the wording of the type of study, in accordance with the limitations of the manuscript. 

It is recommended to include the characteristics of the interviewer: gender, possible biases and interests in the research. 

 It is necessary to include how the sample and participants were selected, how the questionnaires were sent, ----, the reasons for non-participation, ..... 

2.2. Measurement

Although it may be considered a minor issue, it would be advisable to modify the order of presentation of the variables, raising the LH and WE to the first positions, after the socio-demographic variables. This modification indirectly gives them greater weight as dependent variables, on which the objective is focused. 

 It is necessary to include in the description of the variables whether they are dependent or independent.

 3. Results: 

In table 1 it is recommended to include descriptive results for the 8 variables considered. Neither data on presenteeism, nor data on body concern and job satisfaction/family life satisfaction are included. It is necessary to include the descriptive results in the table. 

 Data relating to: Physical activity levels, sleep quality and HL exposed in L186-L189 are included in table 1 It is recommended to remove duplication. 

 In table 2 the data in the first row have been moved, with the column title not corresponding to the data presented. Also the data in the legend is shown out of order Correct for proper display. 

4 Discussion: 

The information contained in L210-212 and L 227-229 is the same. Eliminate duplication. 

Align the limitations with what is stated in section 2.1 Study design, setting, and sample. 

5. Conclusion:  Assess the inclusion regarding HL and bring it in line with the rewritten OBJECTIVE

Author Response

Response to the Reviewer 2

Title: not adequate. It is recommended that the title be rewritten to reflect the two key aspects, work engagement (WE) and Health literacy (HL).

Response

Thank you for your suggestion. Work engagement (WE) is actually a dependent variable in this study, whereas health literacy is an independent variable. We evaluated HL's impact on WE. As a result, we updated the introduction section, and we changed the table title (table 5) in the result section to the impacts.

Abstract: to be rewritten to adapt the suggestions in the manuscript

Response

The authors have revised the abstract accordingly.

- Introduction:
It is brief and adequate, emphasising and defining the essential concepts of the study. However, although the definition made in the introduction of the HL is clear and relevant, the importance and its treatment as a dependent variable in the study is not sufficiently defined
.

Response

The authors have clearly mentioned in the Method section (data analysis part) that the Utrecht Work Engagement Scale (UWES) score is a dependent variable and other items are independent variables.

In L66 mention is made of "the relationship between all these concepts together with HL and WE" and in the results it is presented in section 2.3.3. Relationship between levels of HL and other variables (L158).
Although the value that is intended to be given to HL (at the same level as WE) is intuited in the definition of the objective, HL is diluted, it disappears, there is no mention of it, only of WE,

It is necessary to rewrite the definition of the OBJECTIVE to include this variable (HL).

Response

The authors are extremely sorry for this inconvenience. In this study, HL is the independent variable and WE is the dependent variable, therefore, we have rewritten the characteristics by replacing the impacts of health literacy on other variables among the three companies (Table 5).

  1. Methods:
    2.1 Study design, setting and sample.
    L73 where it says "prospective observational cohort study" it is recommended to review and, if necessary, modify the wording of the type of study, in accordance with the limitations of the manuscript.

Response

The authors have reviewed and revised it accordingly.

It is recommended to include the characteristics of the interviewer: gender, possible biases and interests in the research.

It is necessary to include how the sample and participants were selected, how the questionnaires were sent, ----, the reasons for non-participation, .....

Response

In this research, the authors did not do interviews. It was a self-administered questionnaire; therefore, the questionnaire was delivered to the respondents through email or in-house message. The authors have added these sentences in the method section.

The personnel manager asked the employees about their participation in this study, however, we were unable to know the exact reason for their non-participation in this study.

2.2. Measurement

Although it may be considered a minor issue, it would be advisable to modify the order of presentation of the variables, raising the HL and WE to the first positions, after the socio- demographic variables. This modification indirectly gives them greater weight as dependent variables, on which the objective is focused.

It is necessary to include in the description of the variables whether they are dependent or independent.

Response

The authors already have checked and revised the order of presentation accordingly and clearly mentioned in the method section that WE is a dependent variable and HL is an independent variable.

  1. Results:

In table 1 it is recommended to include descriptive results for the 8 variables considered. Neither data on presenteeism, nor data on body concern and job satisfaction/family life satisfaction are included. It is necessary to include the descriptive results in the table.

Response

We included those variables in previous table 2 which may misinterpret the explanation; therefore, we made a new Table 1 that solely focused on sociodemographic variables and a new Table 2 that shows the Score of the participant scales. We included a new table 3 that includes the Health literacy (HLS-EU-Q47) and Quality of sleep (PSQI).

Data relating to: Physical activity levels, sleep quality, and HL exposed in L186-L189 are included in table 1 It is recommended to remove duplication.

Response

Thank you for your valuable suggestions. The authors have revised it accordingly. The Physical activity level is not a scale and is related to job type, therefore we included this in Table 1.

In table 2 the data in the first row have been moved, with the column title not corresponding to the data presented. Also the data in the legend is shown out of order Correct for proper display.

Response

The authors have revised and corrected it accordingly.

4 Discussion:

The information contained in L210-212 and L 227-229 is the same. Eliminate duplication.

Response

The authors have revised it accordingly and removed the duplication from the result section.

Align the limitations with what is stated in section 2.1 Study design, setting, and sample.

Response

The authors have revised the limitation section aligned with the Study design, setting, and sample.

  1. Conclusion: Assess the inclusion regarding HL and bring it in line with the rewritten

Response

Thank you for your suggestion. We have added a sentence in the conclusion section as suggested.
